# Non-invasive electromyographic speech neuroprosthesis: a geometric perspective

## Abstract

In this article, we present a high-bandwidth *egocentric* neuromuscular speech interface for translating silently voiced speech articulations into text and audio. Specifically, we collect electromyogram (EMG) signals from multiple articulatory sites on the face and neck as individuals articulate speech in an alaryngeal manner to perform EMG-to-text or EMG-to-audio translation. Such an interface is useful for restoring audible speech in individuals who have lost the ability to speak intelligibly due to laryngectomy, neuromuscular disease, stroke, or trauma-induced damage (e.g., radiotherapy toxicity) to speech articulators. Previous works have focused on training text or speech synthesis models using EMG collected during *audible* speech articulations or by transferring audio targets from EMG collected during *audible* articulation to EMG collected during *silent* articulation. However, such paradigms are not suited for individuals who have already lost the ability to *audibly* articulate speech. We are the first to present an alignment-free EMG-to-text and EMG-to-audio conversion using only EMG collected during *silently* articulated speech in an open-sourced manner. On a limited vocabulary corpora, our approach achieves almost $2.4\times$ improvement in word error rate with a model that is $25\times$ smaller by leveraging the inherent geometry of EMG.

## 1. Introduction

Electromyogram (EMG) signals gathered from the orofacial neuromuscular system during the silent articulation of speech in an alaryngeal manner can be synthesized into personalized audible speech, potentially enabling individuals

---

[1]Anonymous Institution, Anonymous City, Anonymous Region, Anonymous Country. Correspondence to: Anonymous Author <anon.email@domain.com>.

*Proceedings of the $42^{nd}$ International Conference on Machine Learning*, Vancouver, Canada. PMLR 267, 2025. Copyright 2025 by the author(s).

without vocal function to communicate naturally. Furthermore, such systems could seamlessly interface with virtual environments where audible communication might disturb others (e.g., multiplayer games) or facilitate telephonic conversations in noisy environments. A key enabler of such advancements is the rich information encoded in EMG signals recorded from multiple spatially separated locations, which capture muscle activation patterns across different muscles. This richness allows for the decoding of subtle and intricate details, such as nuanced speech articulations, likely with higher bandwidth and lower latency compared to exocentric or allocentric modalities, such as video-based lip-to-speech synthesis. By leveraging this information, EMG-based systems offer a promising foundation for natural and efficient communication across diverse applications.

In this article, we present EMG-to-language translation models with a focus on data *geometry*. We show that EMG-to-language translation can be cast as a graph-connectivity learning problem and provide a *single*-layer recurrent architecture with connectionist temporal classification (CTC) loss (Graves et al., 2006) on the manifold of symmetric positive definite (SPD) matrices. Our alignment free translation method is similar to paradigms proposed for invasive speech brain-computer interfaces described by Willett et al. and Metzger et al. While invasive methods are viable for individuals with anarthria or amyotrophic lateral sclerosis, our EMG based non-invasive speech prosthesis is appropriate for individuals who have undergone laryngectomy or experience dysarthria or dysphonia.

On a limited English dataset with a vocabulary of 67 words, we demonstrate that our model achieves a decoding accuracy of 88% on word transcriptions. On a larger, general English language corpus, we achieve a phoneme error rate (PER) of 56%, as measured using Levenshtein distance (between the original and constructed phoneme sequences). Additionally, we show that the model can be trained with minimal data, achieving good performance even when tested on a dataset nearly 5 times larger than the training set, where sentences are spelled out using NATO phonetic codes. This capability is crucial, as collecting large-scale datasets for such systems is often challenging. These results highlight the potential for the practical deployment of such interfaces at scale.

## 2. Prior work

The current benchmark in silent speech interfaces is established by Gaddy & Klein; Gaddy & Klein. Using electromyogram (EMG) signals collected during *silently* articulated speech ($E_S$) and *audibly* articulated speech ($E_A$), along with corresponding audio signals ($A$), they develop a recurrent neural transduction model to map time-aligned features of $E_A$ or $E_S$ with $A$. In their baseline model, joint representations between $E_A$ and $A$ are learned during training, and the model is tested on $E_S$. To improve performance, a refined model aligns $E_S$ with $E_A$ and subsequently uses the aligned features to learn joint representations with $A$. The methods described above have significant shortcomings that limit their practicality for real-world deployment. They are, ① the unavailability of good quality $E_A$ and $A$ in individuals who have lost vocal and articulatory functions, ② the need for a *2x* sized training corpus for learning *x* representations (both $E_A$ and $E_S$), and ③ the requirement of aligned features, which are computationally expensive and time-consuming to obtain, making near real-time implementation challenging. We overcome the above challenges by training a model with only $E_S$ and corresponding phonetic transcription without any alignments, using CTC loss. Besides, unlike Gaddy & Klein; Gaddy & Klein, we demonstrate the efficacy of our models on multiple subjects.

Another notable approach is presented by Gowda et al., who demonstrate that, unlike images and audio - which are functions sampled on Euclidean grids - EMG signals are defined by a set of orthogonal axes, with the manifold of SPD matrices as their natural embedding space. We build upon the methods described by Gowda et al. in our analysis and introduce the following key improvements: ① we train a recurrent model for EMG-to-phoneme sequence-to-sequence generation, as opposed to the classification models proposed by Gowda et al., ② we operate in the sparse graph spectral domain, effectively circumventing bottlenecks associated with repeated eigenvalue computation in neural networks, which, due to their iterative nature, often have limited parallelization capabilities on GPUs, and ③ demonstrate EMG-to-language conversion on continuously articulated speech as opposed to individual words or phonemes.

A substantial body of prior work (Jou et al., Schultz & Wand, Kapur et al., Meltzner et al., Toth et al., Janke & Diener, and Diener et al.) has laid the groundwork for the development of silent speech interfaces. While these studies have been instrumental in shaping the field, they place less emphasis on understanding the *data structure* and the implementation of parameter and data-efficient approaches.

In the following sections, ① we explain the inherent non-Euclidean data structure of EMG signals, ② quantify the signal distribution shift across individuals, and ③ demonstrate that high fidelity phoneme-by-phoneme translation of

EMG-to-language is possible using only $E_S$ without $E_A$ and $A$.

## 3. Methods

EMG signals are collected by a set of sensors $\mathcal{V}$ and are functions of time $t$. A sequence of EMG signals $E_S$ corresponding to silently articulated speech, associated with audio $A$ and phonemic content $L$, is represented as $E_S = \mathbf{f}_v(t)$ for all $v \in \mathcal{V}$. Here, $\mathbf{f}_v(t)$ denotes the EMG signal captured at a sensor node $v$ as a function of time $t$. The audio signal $A$ encodes both phonemic (lexical) content and expressive aspects of speech, such as volume, pitch, prosody, and intonation, while $L$ represents purely the phonemic content - a sequence of phonemes. For instance, the phonemic content $L$ of the word $<$FRIDAY$>$ is denoted by the phoneme sequence $<$F-R-AY-D-IY$>$.

To model the mapping from $E_S$ to $L$, we employ a sequence-to-sequence model trained using CTC loss. This approach allows us to train the model with *unaligned* pairs of $E_S$ and $L$, eliminating the need for precise alignment between the input signals and their corresponding phoneme sequences. During testing, a sample of $E_S$ not in the training set outputs probabilities over all possible phonemes (40 of them in our case) at every time step, and we construct $L$ using beam search. $L$ is then converted to personalized audio $A$ using few-shot learning (Choi et al., 2021), which requires as little as a single audio clip from the individual (an audio clip of about 3-5 minutes, not necessarily containing the same phonemic content as $L$, recorded before their clinical condition). By leveraging this sample, we generate audio $A$ that captures both the predicted linguistic content and the speaker's unique vocal characteristics (we elaborate on this topic in appendix G).

### 3.1. EMG data representation

Gowda et al. demonstrate that the manifold of SPD matrices serves as an effective embedding space for EMG signals, enabling the natural distinction of different orofacial movements associated with speech articulation and all English phonemes using raw signals. We make significant improvements on their methods to perform phoneme-by-phoneme decoding as opposed to classification paradigms and demonstrate our methods on continuously articulated speech in the English language as opposed to discrete word or phoneme articulations.

We construct a complete graph $\mathcal{G} = (\mathcal{V}, \mathcal{E}(\tau))$ to represent the functional connectivity of EMG signals, where $\mathcal{E}(\tau)$ denotes the set of edges over a time window $\tau = [t_{\text{START}}, t_{\text{END}}]$ (Gowda et al., 2024). The edge weight between two nodes $v_1$ and $v_2 \in \mathcal{V}$ in a time window is defined as $e_{12} = e_{21} = \mathbf{f}_{v_1}^T \mathbf{f}_{v_2}$, which corresponds to the covari-

ance of the signals at those nodes during the time interval. Consequently, the edge (adjacency) matrix $\mathcal{E}(\tau)$ is symmetric and positive semi-definite. Following Gowda et al., we convert semi-definite adjacency matrices into definite ones by adding a shrinkage estimator. We then model these symmetric positive definite (SPD) matrices using the Riemannian geometry approach via Cholesky decomposition, as described by Lin.

For any adjacency matrix $\mathcal{E}$, we can express it as $\mathcal{E} = U\Sigma U^T$, where $U$ is the matrix of eigenvectors, and $\Sigma$ is a diagonal matrix containing the corresponding eigenvalues. However, instead of calculating $U$ for each $\mathcal{E}$ at every timestep $\tau$, we fix an approximate common eigenbasis $Q$ derived from the Fréchet mean $\mathcal{F}$ (Lin, 2019) of all adjacency matrices (at different time points) in the training set. Specifically, we compute $\mathcal{F}$ as the geometric mean of all $\mathcal{E}$, and decompose it as $\mathcal{F} = Q\Lambda Q^T$, where $Q$ contains the eigenvectors of $\mathcal{F}$, and $\Lambda$ is a diagonal matrix of its eigenvalues.

Using this fixed eigenbasis $Q$, any adjacency matrix $\mathcal{E}$ can be approximately diagonalized as $Q^T\mathcal{E}Q$, yielding a sparse matrix $\sigma$. Gowda et al. show that such a matrix $Q$ can be learned using neural networks constrained on the Stiefel manifold (Huang & Van Gool, 2017) and that such a $Q$ is different for different individuals. However, neural networks constrained on the Stiefel manifold require performing repeated eigendecomposition operations, which have limited parallelization capability and lead to unstable gradients while using CTC loss. Therefore, we simply derive $Q$ from the Fréchet mean $\mathcal{F}$ and use that $Q$ to obtain sparse matrices $\sigma$. In appendix D, we show that matrices $\sigma$ are indeed sparse by comparing the ratio of maximum value of non-diagonal entries to maximum value of diagonal entries of matrices before and after approximate diagonalization. This formulation allows us to work in an approximate graph spectral domain with a consistent orthogonal basis across all time windows $\tau$. For our task, we compute the graph spectral sequences $\sigma$ for all time windows $\tau$ and use these as inputs for EMG-to-language translation. We illustrate these concepts in figure 1.

### 3.2. Sequence-to-sequence modeling with a single recurrent layer

We implement a single-layer gated recurrent unit (GRU) architecture for EMG-to-phoneme sequence-to-sequence modeling. The input to the GRU consists of a sequence of approximately diagonalized matrices, denoted as $\sigma$, derived over different time windows $\tau$.

To investigate whether recurrent models defined on the manifold provide a better representation of $\sigma(\tau)$ compared to those defined in Euclidean space, we construct three distinct GRU architectures:

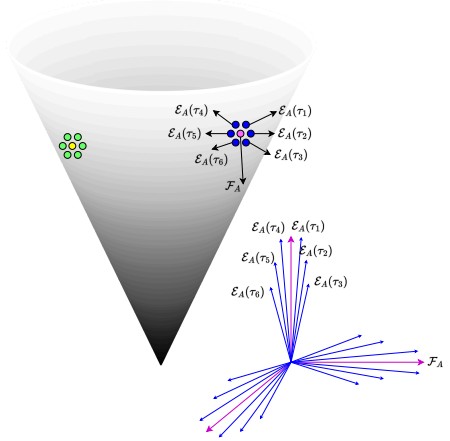

*Figure 1.* Conceptual depiction of SPD edge matrices on a 3D convex cone manifold. Edge matrices derived from a given individual can be represented using an approximate *common eigenbasis*. Consider six edge matrices, corresponding to six different time windows $\tau_1$ to $\tau_6$, for an individual $A$. These matrices are shown in *blue*. The Fréchet mean of these SPD matrices, denoted as $\mathcal{F}_A$, is represented in *pink*. Each edge matrix $\mathcal{E}_A(\tau_i)$ can be expressed as: $\mathcal{E}_A(\tau_i) = U_{A_i}\Sigma_{A_i}U_{A_i}^T$, where $\Sigma_{A_i}$ is a diagonal matrix of eigenvalues, and $U_{A_i}$ is an orthogonal matrix of eigenvectors ($i \in [1,6]$). Instead of representing each edge matrix with a separate eigenvector matrix, we transform all $\mathcal{E}_A(\tau_i)$ using a *common basis* $Q_A$, which corresponds to the eigenvectors of the Fréchet mean $\mathcal{F}_A$. Specifically, we calculate $\sigma_{A_i} = Q_A^T\mathcal{E}_A(\tau_i)Q_A$. Matrices $\sigma_{A_i}$ are approximately diagonal. We use these approximately diagonalized representations for EMG-to-language translation. Separately, edge matrices from another individual $B$, shown in *green*, have their Fréchet mean represented in *yellow*. Matrices belonging to different individuals reside in distinct neighborhoods on the manifold, and the common basis $Q_A$ for individual $A$ cannot approximately diagonalize edge matrices from individual $B$. Instead, a separate basis $Q_B$, derived from the eigenvectors of $\mathcal{F}_B$, is required for individual $B$. Geometrically, the manifold is only *locally Euclidean* and the tangent spaces for individuals $A$ and $B$ are distinct. That is, transformation of the space $\mathbb{R}^{|\mathcal{V}|}$ induced by EMG signals of subjects $A$ and $B$ are different and the approximate orthogonal eigenbasis vectors that characterize such transformations are different for different individuals. This signal distribution shift can be approximated as change of basis. An inset diagram illustrates the eigenvectors of $\mathcal{E}_A(\tau_i)$.

① $\textbf{GRU}_A$: A GRU layer defined in the Euclidean domain, following the implementation described by Chung et al. (2014),

② $\textbf{GRU}_B$: A GRU layer formulated on the manifold of SPD matrices, as proposed by Jeong et al. (2024), and

③ $\textbf{GRU}_C$: A GRU layer defined on the manifold of SPD matrices, plus an implicit layer solved using neural ordinary differential equations, integrating methodologies from Jeong et al. (2024), Chen et al. (2018), and Lou et al. (2020).

GRU$_B$ and GRU$_C$ directly accept SPD matrices, $\sigma$, as input, whereas GRU$_A$ processes the vectorized representations of $\sigma$. At each time step, the GRU models output probability distributions over 40 phonemes in the English language. The models are trained using CTC loss, and during inference, the most probable phoneme sequence is reconstructed using beam search decoding. The end-to-end EMG-to-language translation model is depicted in figure 2.

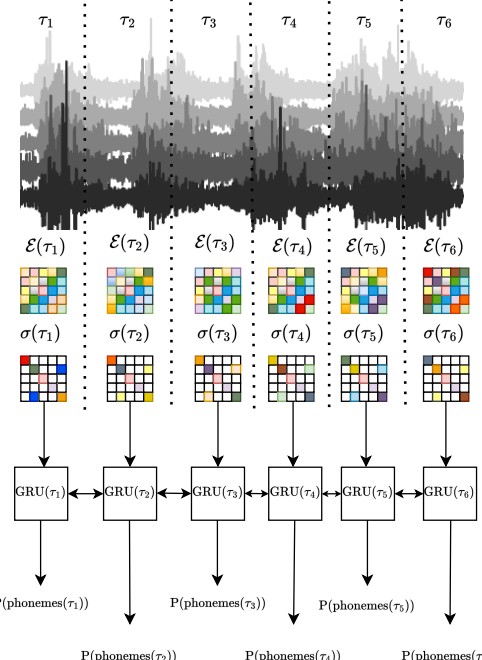

*Figure 2.* Illustration of *multivariate EMG-to-phoneme sequence translation*. Bandpass-filtered and $z$-normalized raw signals are converted to SPD edge matrices, $\mathcal{E}(\tau)$, over a time window $\tau$. These edge matrices are then transformed into approximately diagonalized matrices, $\sigma(\tau)$, which are fed into a bidirectional GRU layer. At each time step (every 20 ms), the GRU outputs probability distributions $P$ over 40 phonemes in the English language. During inference, the most probable phoneme sequence is reconstructed using beam search decoding.

### 3.3. Geometric perspective aligns well with biology

We study multivariate EMG signals collected at $|\mathcal{V}|$ sensor nodes in different time windows $\tau$ using edge matrices $\mathcal{E}(\tau)$, which capture the relationship between every pair of nodes in $|\mathcal{V}|$.

This can be understood as studying the transformation of the space $\mathbb{R}^{|\mathcal{V}|}$ given by the transformation matrix $\mathcal{E}(\tau)$. Such a transformation can equivalently be expressed in a coordinate system where the eigenvectors serve as the basis vectors. This change of basis is described by $U^T \mathcal{E}(\tau) U = \Sigma(\tau)$, where $\Sigma(\tau)$ is a diagonal matrix. In this eigenbasis coordinate system, transformation of space induced by EMG

signals can be interpreted as a linear combination of the columns of $U$, with the diagonal values of $\Sigma$ acting as coefficients. By fixing an approximate eigenbasis $Q$, we obtain an approximately diagonal matrix $\sigma(\tau)$ and an approximate linear combination. This formulation aligns well with the biological process underlying EMG signal generation, which involves a purely additive combination of muscle activations which contrasts with processes such as speech production, which can be modeled as the application of a time-varying filter to a time-varying source signal (Sivakumar et al.). For a given individual, we fix the approximate eigenbasis vectors and focus on analyzing the approximate eigenvalues $\sigma(\tau)$. These matrices can then be studied using a single layer recurrent neural network. EMG signals from different individuals induce different transformations of $\mathbb{R}^{|\mathcal{V}|}$ and have different eigenbasis vectors. The signal distribution shift across different individuals can thus be interpreted as change of basis in $\mathbb{R}^{|\mathcal{V}|}$. It should be noted that while a *shallow* single layer network is sufficient to learn multivariate EMG-phoneme translation using sparse matrices $\sigma$, and while such an architecture works consistently well across subjects, the weights of the recurrent networks must be fine-tuned for different individuals, as $\sigma$ from different individuals correspond to different basis vectors $Q$.

## 4. Data

We evaluate our models using three datasets, referred to as Data$_{\text{SMALL-VOCAB}}$, Data$_{\text{LARGE-VOCAB}}$, and Data$_{\text{NATO-WORDS}}$, which are described below. We use Data$_{\text{SMALL-VOCAB}}$ and Data$_{\text{LARGE-VOCAB}}$ to evaluate naturally articulated speech in a silent manner. We use Data$_{\text{NATO-WORDS}}$ to demonstrate that, by using a small codeword set such as NATO codes, we can construct a generalizable language-spelling model that requires very little data for training. Additionally, Data$_{\text{NATO-WORDS}}$ is used to show that our models work consistently well across individuals. This paradigm is useful for rapid training (or fine-tuning) and deployment of speech prostheses.

### 4.1. Data$_{\text{SMALL-VOCAB}}$

Following Gaddy & Klein, we create a limited vocabulary dataset consisting of 67 unique words. These words include weekdays, ordinal dates, months, and years. Sentences are constructed from these words in the format <WEEKDAY-MONTH-DATE-YEAR>. A single individual articulated 500 such sentences silently, and the resulting EMG data, $E_S$, is translated into output phoneme sequences. We have timestamps to demarcate the beginning and the end of words within a sentence.

We collect EMG data from 31 muscle sites at a sampling rate of 5000 Hz. Of these, 22 electrode sites are identical to those used by Gowda et al., while the remaining 9 electrodes

are placed symmetrically on the opposite side of the neck. The experimental setup is same as that described by Gowda et al.

### 4.2. Data LARGE-VOCAB

We adapt the language corpora from Willett et al., who demonstrated a speech brain-computer interface by translating neural spikes from the motor cortex into speech. The dataset comprises an extensive English language corpus containing approximately 6,500 unique words and 11,000 sentences. Unlike Gaddy & Klein; Gaddy & Klein, we collect only $E_S$ (excluding $E_A$ and $A$) and perform $E_S$-to-language translation without time-aligning with $E_A$ and $A$. The data collection setup follows the methodology described for Data SMALL-VOCAB. This corpus includes sentences of varying lengths, with the subject articulating sentences at a normal speed, averaging 160 words per minute. Timestamps were used solely to mark the beginning and end of each sentence, with the subject clicking the mouse at the start of articulation and again upon completion (unlike Data SMALL-VOCAB, there are no timestamps to demarcate between words within a sentence).

### 4.3. Data NATO-WORDS

We use the dataset provided by Gowda et al.[1] Specifically, we use data from their second experiment, in which 4 individuals articulated English sentences in a spelled-out manner using NATO phonemic codes in a silent manner. For instance, the word <RAINBOW> was articulated as <ROMEO-ALFA-INDIA-NOVEMBER-BRAVO-OSCAR-WHISKEY> with phonemic transcription <R-OW-M-IY-OW|AE-L-F-AH | IH-N-D-IY-AH | N-OW-V-EH-M-B-ER | B-R-AA-V-OW | AO-S-K-ER | W-IH-S-K-IY>. Subjects articulated phonemically balanced RAINBOW and GRAND-FATHER passages in this spelled-out format. In total, 1968 NATO code articulations were recorded across both passages.

The EMG data was collected from 22 muscle sites in the neck and cheek regions at a sampling rate of 5000 Hz. We present results for Data NATO-WORDS in appendix A.

## 5. Results

We describe the experimental setup and results for Data SMALL-VOCAB and Data LARGE-VOCAB, providing a comparative analysis with previous benchmarks.

### 5.1. Results for Data SMALL-VOCAB

Raw EMG signals are bandpass filtered between 80 and 1000 Hz and are *z*-normalized per channel along the time

---

[1] The dataset is available at Gowda et al. dataset.

dimension. Then, a complete time dependent graph ($\mathcal{E}(\tau)$ and $\sigma(\tau)$) is constructed using the EMG signals. We follow the same *train-validation-test* split outlined by Gaddy & Klein. All parameters are detailed below in table 1.

*Table 1.* Experimental setup for Data SMALL-VOCAB.

| | **Data$_A$ properties** |
|---|---|
| $\tau$ | 50 ms (a sliding window with an overlapping context size of 100 ms and a step size of 50 ms) |
| $\mathcal{E}(\tau)$ and $\sigma(\tau)$ | SPD matrices of dimensions $31 \times 31$ |
| Train-validation-test split | 370 - 30 - 100 sentences |
| Beamsearch width | Top-*5* |

The Fréchet mean, computed from the training set, is utilized to calculate $\sigma(\tau)$ for all $\tau$ in the training, validation, and test datasets. Figure 3 illustrates the Levenshtein distances between target and predicted phoneme sequences for three GRU models with varying model sizes. To decode the articulated word(s), we identify a word or a set of words from the vocabulary corpus whose phonemic sequence best matches the predicted sequence, using Levenshtein distance as the metric.

Decoding accuracy for EMG-to-text translation is evaluated as $1 - $ WER and is presented in figure 4 for models of different sizes. Model size is controlled exclusively by adjusting the GRU hidden unit dimensionality, which is the ***only hyperparameter*** in our setup. On this limited vocabulary corpus, we achieve a WER as low as 12%, with the average Levenshtein distance between target and predicted sequences below 1. These results underscore the feasibility and practical potential of EMG-to-language translation technology.

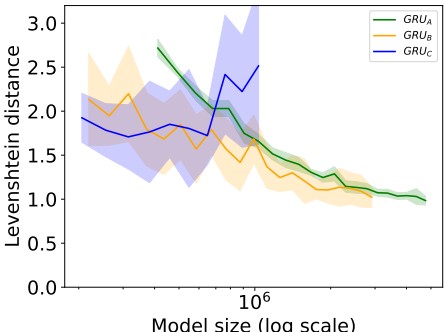

*Figure 3.* Model size versus Levenshtein distance. Models are evaluated over 10 random seeds. *Lower is better.*

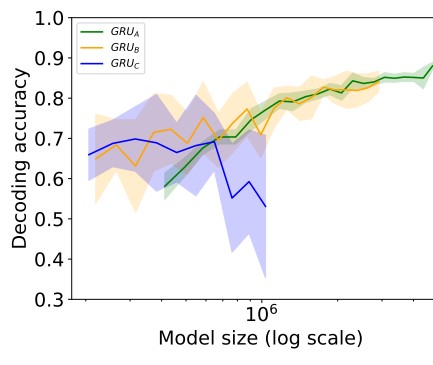

*Figure 4.* Decoding accuracy = 1 - WER versus model size. Models are evaluated over 10 random seeds. *Higher is better.*

Now, we compare our results with the results given by Gaddy & Klein. Gaddy & Klein recorded EMG signals using 8 electrodes at a sampling rate of 1000 Hz. To enable a direct comparison with their approach, we downsample our EMG signals from 5000 Hz to 1000 Hz and select a subset of 8 electrodes from the original 31. The placement of these electrodes is approximately aligned with those specified by Gaddy & Klein to ensure consistency in the experimental setup.

We compare our results with their baseline approach, where models were trained for $E_A$-to-$A$ translation and evaluated on $E_S$-to-$A$ translation. In contrast, our approach emphasizes direct $E_S$-to-$L$ translation. Additionally, their improved framework incorporates both $E_A$ and $E_S$ signals, relying on time alignment between them. However, they do not propose a paradigm that independently translates $E_S$ without leveraging $E_A$ or $A$. In this case, $\mathcal{E}(\tau)$ and $\sigma(\tau)$ are 8×8 matrices. The rest of the training paradigm is same as in table 1. We provide the comparison in table 2. Our approach achieves almost $2.4\times$ improvement in WER with a model that is $25\times$ smaller.

*Table 2.* Comparison with Gaddy & Klein. Our approach achieves almost $2.4\times$ improvement in WER with a model that is $25\times$ smaller. WER is averaged over 10 random seeds. Results are for Data$_{\text{SMALL-VOCAB}}$ using 8 electrodes with signals downsampled to 1000 Hz.

| **Ours** | Baseline of Gaddy & Klein |
|---|---|
| **WER - 27%**, using GRU$_B$ | WER - 64% |
| Model size - about **1.4 million** | Model size - about 40 million |

## 5.2. Results for Data$_{\text{LARGE-VOCAB}}$

As in Data$_{\text{SMALL-VOCAB}}$, we filter, $z$-normalize, and construct $\sigma(\tau)$. The properties of the dataset are detailed in table 3. Sentences in the validation and test sets do not occur in the training set. On this general English language corpus consisting of approximately 6500 words, spoken at an average rate of 160 words per minute, we perform EMG-to-phoneme sequence translation and measure the Levenshtein distances between the target and predicted phoneme sequences. The phoneme error rates (PER) are presented in table 4. Transcription examples are given in table 5.

*Table 3.* Experimental setup for Data$_{\text{LARGE-VOCAB}}$.

| | **Data$_{\text{LARGE-VOCAB}}$ properties** |
|---|---|
| $\tau$ | 20 ms (a sliding window with an overlapping context size of 50 ms and a step size of 20 ms) |
| $\mathcal{E}(\tau)$ and $\sigma(\tau)$ | SPD matrices of dimensions $31\times 31$ |
| Train-validation-test split | 8000 - 1000 - 1970 sentences |
| Beamsearch width | Top-5 |

*Table 4.* We achieve a PER of 56% for speech articulated at 160 words per minute using 31 electrodes. Average phoneme sequence length of sentences is 24.5, and the chance decoding accuracy of a sequence (that is, chance 1 - PER) is $\left(\frac{1}{40}\right)^{24.5}$. While the results are modest compared to high density invasive speech brain-computer interfaces, we showcase significant potential of a non-invasive method. Willett et al. report a PER of 21% using 128 intracortical arrays at a slower speech rate of 62 words per minute. Metzger et al. report a PER of 30% on a smaller corpora of 1024 words articulated at a slower rate of 78 words per minute using 253 ECoG electrodes. In future work, we would like to verify if higher density EMG and more training data can lead to better PER.

| Ours - 31 electrodes |
|---|
| PER - 56%, using GRU$_A$ |
| Model size - about 4.4 million |

## 6. Observations and discussions

① From Data$_{\text{SMALL-VOCAB}}$ and Data$_{\text{LARGE-VOCAB}}$ and their corresponding results, we observe that continuously articulated silent speech - where subjects naturally articulate sentences in English but inaudibly - can be translated into phonemic sequences at a fine-scale resolution of 50 ms or 20 ms. This resolution is comparable to state-of-the-art (SOTA)

automatic speech recognition (ASR) models, such as those described by Baevski et al. and Hsu et al., which operate at a 20 ms resolution. These findings highlight the potential for real-time EMG-to-language translation, akin to audio-to-audio (language translation) and audio-to-text translation. Furthermore, achieving a word transcription accuracy of approximately 88% on limited-vocabulary corpora and a PER of 56% on open-vocabulary corpora demonstrates that high-fidelity translation is achievable, reinforcing the viability of this approach.

② Additionally, results from Data $_{\text{NATO-WORDS}}$ (WER of 59% averaged across all 4 subjects) indicate that by leveraging NATO phonetic codes, we can establish a generalizable EMG-to-language spelling paradigm. Although this approach does not replicate natural speech, it enables a practical mode of limited communication for individuals who have lost speech articulation capabilities. Notably, this paradigm is efficient, requiring only a small corpus for training - our model trained on just *10 minutes* of data, demonstrates robust generalization on a much larger test set.

③ The key contribution of this article lies in the development of efficient architectures for multivariate EMG-to-phoneme sequence translation. We show that EMG signals can be approximately decomposed into linear combinations of a set of orthogonal axes, represented by the matrices $\sigma(\tau)$. This decomposition enables the analysis of time-varying graph edges in a sparse graph spectral domain using a single recurrent layer. Notably, our model relies on only one **hyperparameter** - the dimension of the hidden unit in the GRU. Across different datasets and subjects, the models exhibit consistent and predictable behavior with respect to the hidden unit dimension. Specifically, the decoding accuracy of GRU$_A$ and GRU$_B$ improves as the hidden unit dimension increases, eventually plateauing, while GRU$_C$ demonstrates a peak in performance before diminishing (figures 4 and 6). Also, GRU$_C$ outperforms other GRU models for Data $_{\text{NATO-WORDS}}$. It also outperforms other GRU models for Data $_{\text{SMALL-VOCAB}}$ at smaller model sizes. This demonstrates that modeling dynamics of EMG signals using neural ODEs is beneficial and allows for better abstraction of the data. Importantly, although different datasets and individuals are characterized by distinct orthogonal basis vectors, the same model architecture can be applied across individuals without the need for extensive hyperparameter tuning.

④ We achieve a word error rate (WER) of approximately 12% on a 67-word vocabulary, not too far from the 9.1% WER reported by Willett et al. on a 50-word vocabulary. Notably, our results are achieved using only 31 non-invasive electrodes, in contrast to the 128 intracortical electrodes employed by Willett et al. On Data $_{\text{LARGE-VOCAB}}$, we achieve a phoneme error rate (PER) of 56% for speech articulated

at an average rate of 160 words per minute, whereas Willett et al. report a PER of 21% using 128 intracortical arrays at a slower speech rate of 62 words per minute. In future work, we would like to verify if higher density EMG and more training data can lead to better PER. These findings demonstrate the feasibility of a non-invasive approach for translating silent speech into language. While Willett et al. and Metzger et al. showcase brain-computer speech interfaces for individuals with anarthria or amyotrophic lateral sclerosis, our method provides a viable alternative for individuals who have undergone laryngectomy or experience dysarthria or dysphonia, where invasive recordings may not be a practical solution. We highlight the significant potential of non-invasive techniques for broad clinical applicability.

⑤ Défossez et al. demonstrate methods for decoding speech perception from non-invasive neural recordings using magnetoencephalography (MEG) and electroencephalography (EEG). They show that *listened* speech segments can be predicted from MEG with an accuracy of 41%. However, such interfaces are not useful for initiating communication. We go beyond these models to demonstrate that, using non-invasive EMG signals, we can decode speech articulation at the phonemic level with a higher accuracy of 44% on a large English language corpus.

⑥ We envision EMG-based non-invasive neuroprostheses having a user-friendly form factor that is easy to don and doff. However, even minor variations in electrode placement introduce a covariate signal shift, which can be mathematically represented as a change of basis (Gowda et al., 2024). Additional factors, such as variations in subcutaneous fat and changes in neural drive characteristics, contribute to covariate signal drift over time. To address these challenges, modeling EMG signals using SPD covariance matrices proves advantageous. Our models show consistent performance across subjects, as demonstrated here, and outperform Euclidean-space models (Gaddy & Klein, Gaddy & Klein) in terms of both decoding accuracy and model parameter efficiency. Moreover, considering the idiosyncrasies of individuals, the difficulty of collecting large-scale data, and the need for frequent fine-tuning due to circumstantial variations, a streamlined approach is crucial. A simple model leveraging a single GRU layer, as presented here, offers an effective solution for adaptability.

## 7. Conclusion

We present an efficient data representation for orofacial EMG signals and demonstrate that our approach enables effective EMG-to-language translation. Our method outperforms previous benchmarks on limited-vocabulary corpora, showcasing its potential for practical applications. Notably, we demonstrate the ability to translate EMG collected during *silently* voiced speech ($E_S$) to language without requiring

corresponding audio ($A$) and EMG collected during *audibly voiced speech* ($E_A$), marking a significant advancement in the translation paradigm and paving the way for real-world deployment of such devices. By providing open-source data and code, this work lays a solid foundation for the development of efficient neuromuscular speech prostheses.

In future work, we plan to augment our methods with language models and test their applicability for individuals with clinical etiologies that affect voicing and articulator movement in real time.

*Table 5.* Examples of EMG-to-phoneme sequence translations. We do translations using EMG collected during *silent* articulations ($E_S$) with CTC loss without making use of corresponding time aligned *audio* ($A$) and EMG collected during *audible* articulation ($E_A$). Ground truth sentences with corresponding timestamps. Ground truth phonemic transcriptions. Decoded phonemic transcriptions.

---

**3 transcribed sentences in Data** SMALL-VOCAB

---

T-START <WEDNESDAY>T-END T-START <JULY>T-END T-START <TWENTY SIXTH>T-END T-START <NINETEEN SIXTY SEVEN>T-END

W-EH-N-Z-D-IY SPACE J-UW-L-AY SPACE T-W-EH-N-T-IY-S-IH-K-S-TH SPACE N-AY-N-T-IY-N-S-IH-K-S-T-IY-S-EH-V-AH-N

W-AH-N-Z-D-IY SPACE J-UW-L-AY SPACE T-W-EH-N-T-IY-S-IH-K-S-TH SPACE N-AY-N-T-IY-N-S-IH-K-S-T-IY-S-EH-V-AH-N

---

T-START <THURSDAY>T-END T-START <OCTOBER>T-END T-START <TWENTY NINTH>T-END T-START <TWO THOUSAND NINE>T-END

TH-ER-Z-D-EY SPACE AA-K-T-OW-B-ER SPACE T-W-EH-N-T-IY-N-AY-N-TH SPACE T-UW-TH-AW-Z-AH-N-D-N-AY-N

TH-ER-Z-D-EY SPACE AA-K-T-OW-B-ER SPACE T-W-EH-N-T-IY-N-AY-N-TH SPACE T-UW-TH-AW-Z-AH-N-D-T-N-AY-N

---

T-START <TUESDAY>T-END T-START <DECEMBER>T-END T-START <FIFTH>T-END T-START <NINETEEN SEVENTY EIGHT>T-END

T-UW-Z-D-IY SPACE D-IH-S-EH-M-B-ER SPACE F-IH-F-TH SPACE N-AY-N-T-IY-N-S-EH-V-AH-N-T-IY-EY-T

T-UW-Z-D-IY SPACE D-IH-S-EH-M-B-ER SPACE F-IH-F-TH SPACE N-AY-N-T-IY-N-S-EH-V-AH-N-T-IY-AY-N-T

---

**Top-3 (best) transcribed sentences in Data** LARGE-VOCAB

---

T-START <ITS KIND OF FUN>T-END

IH-T-S SPACE K-AY-N-D SPACE AH-V SPACE F-AH-N

IH-T-S SPACE K-AY-N-D SPACE F-AH-N

---

T-START <PROBABLY SEVENTIES>T-END

P-R-AA-B-AH-B-L-IY SPACE S-EH-V-AH-N-T-IY-Z

P-R-AH-B-L-IY SPACE S-EH-V-AH-N-T-IY

---

T-START <IS IT LEGEND>T-END

IH-Z SPACE IH-T SPACE L-EH-JH-AH-N-D

IH-T SPACE IH-T SPACE S-EH-JH-AH-N

---

**Bottom-3 (worst) transcribed sentences in Data** LARGE-VOCAB

---

T-START <MEMBER OF THE AMERICAN METEOROLOGICAL SOCIETY>T-END

M-EH-M-B-ER SPACE AH-V SPACE DH-AH SPACE AH-M-EH-R-AH-K-AH-N SPACE

M-IY-T-IY-AO-R-AH-L-AA-JH-IH-K-AH-L SPACE S-AH-S-AY-AH-T-IY

DH-AH-M-AH SPACE F-AH-B-AE-T-AH SPACE UW SPACE K-L SPACE S-AH SPACE T-IY SPACE D-IH

---

T-START <PICTURES AND PROJECTS THAT YOU CAN MAKE YOURSELF>T-END

P-IH-K-CH-ER-Z SPACE AH-N-D SPACE P-R-AA-JH-EH-K-T-S SPACE DH-AE-T SPACE Y-UW SPACE

K-AE-N SPACE M-EY-K SPACE Y-ER-S-EH-L-F

AH SPACE P-R-IY SPACE T-ER-S-AH SPACE P-R-AA SPACE DH-IH-K SPACE DH-AH SPACE T-UW-K-AH SPACE M SPACE Y-UW

---

T-START <HE EXPECTED CONCLUSIONS AT THE END OF THE YEAR>T-END

HH-IY SPACE IH-K-S-P-EH-K-T-AH-D SPACE K-AH-N-K-L-UW-ZH-AH-N-Z SPACE AE-T SPACE

DH-AH SPACE EH-N-D SPACE AH-V SPACE DH-AH SPACE Y-IH-R

G-EH-P-IH SPACE AH-K-L-UW-ZH-AH SPACE AH-N SPACE AY-D SPACE AH SPACE TH-Y

---

## Impact statement

This article provides data and methods for developing non-invasive EMG based neuroprosthesis. Such devices have the potential to restore natural communication in individuals who have lost the ability to speak intelligibly due to causes such as neuromuscular disease, stroke, trauma, and head/neck cancer surgery (e.g. laryngectomy) or treatment (e.g. radiotherapy toxicity to the speech articulators).

IRB was approved for human subject research. IRB details and its terms and conditions will be made available if the manuscript is accepted.

We are unable to publish the data and codes in an anonymized format. Data and codes will be made publicly available if the manuscript is accepted.

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

## A. Results for Data $_{\text{NATO-WORDS}}$

Raw EMG signals are bandpass filtered between 80 and 1000 Hertz and are *z*-normalized per channel along the time dimension. Then, a complete time dependent graph is constructed using the EMG signals. We follow the same *train-validation-test* split outlined by Gowda et al. All parameters are detailed in table 6.

Like before, the Fréchet mean, computed from the training set, is utilized to calculate $\sigma(\tau)$ for all $\tau$ in the training, validation, and test datasets. Figure 5 illustrates the Levenshtein distances between target and predicted phoneme sequences for three GRU models with varying model sizes across all 4 individuals. To decode the articulated NATO phonetic code, we identify a code from the corpus of 26 codes whose phonetic sequence best matches the predicted sequence, using Levenshtein distance as the metric. Decoding accuracy for EMG-to-text translation is evaluated as $1 - \text{WER}$ and is presented in figure 6 for models of different sizes across all four subjects. For each subject, the best decoding accuracy across all three GRU models and all model sizes are summarized in table 7.

*Table 6.* Experimental setup for Data $_{\text{NATO-WORDS}}$.

| | Data $_{\text{NATO-WORDS}}$ **properties** |
|---|---|
| $\tau$ | 30 ms (a sliding window with an overlapping context size of 150 ms and a step size of 30 ms). |
| $\mathcal{E}(\tau)$ and $\sigma(\tau)$ | SPD matrices of dimensions 22× 22. |
| Train set | 416 NATO alphabet articulations (26 words, each repeated 16 times) |
| Validation set | 104 NATO alphabet articulations (26 words, each repeated 4 times) |
| Test set | 1968 NATO alphabet articulations (entire GRANDFATHER and RAINBOW passages articulated in a spelled-out manner). |
| Beamsearch width | Top-5 |

*Table 7.* Best decoding accuracy across all GRU models and model sizes of all four subjects (calculated by averaging over 10 random seeds). Random chance accuracy is only ***3.85%***.

| **Subject** | Decoding accuracy (1 - WER) via *phoneme-by-phoneme reconstruction* | Gowda et al. classification model accuracy |
|---|---|---|
| 1 | 44.29% | 51.34% |
| 2 | 44.97% | 42.89% |
| 3 | 29.58% | 37.21% |
| 4 | 43.59% | 42.79% |
| **Average** | **40.61%** | **43.56%** |

We compare our results obtained by constructing the most probable phoneme sequences using beam search over the output probability distributions at every time step to that of classification models presented by Gowda et al. in table 7. We see a slight decrease in decoding accuracy. This might be due to the fact that these are single word articulations and classification models have the context of the articulation duration of the entire word as opposed to 150 ms context size in phoneme-by-phoneme sequence-to-sequence modeling.

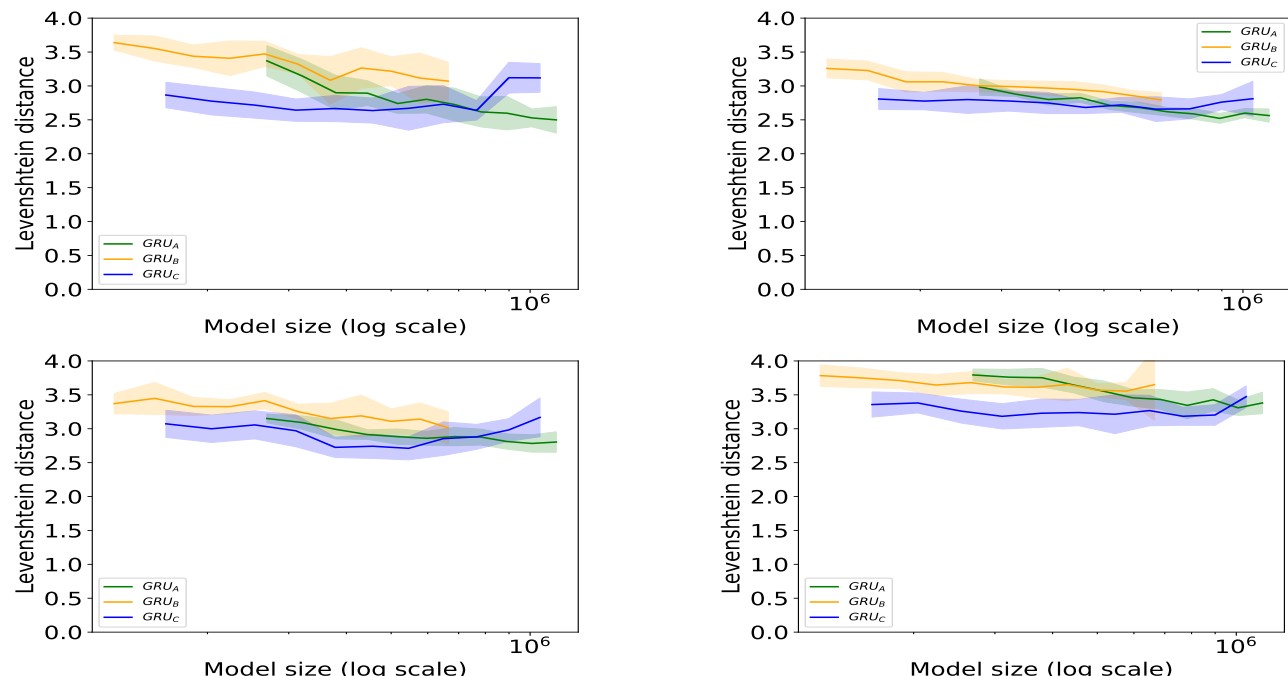

*Figure 5.* Average Levenshtein distance between target and predicted phoneme sequences of all four subjects (subject 1 to subject 4, starting from top left in a clockwise manner). Models are evaluated over 10 random seeds. *Lower is better.*

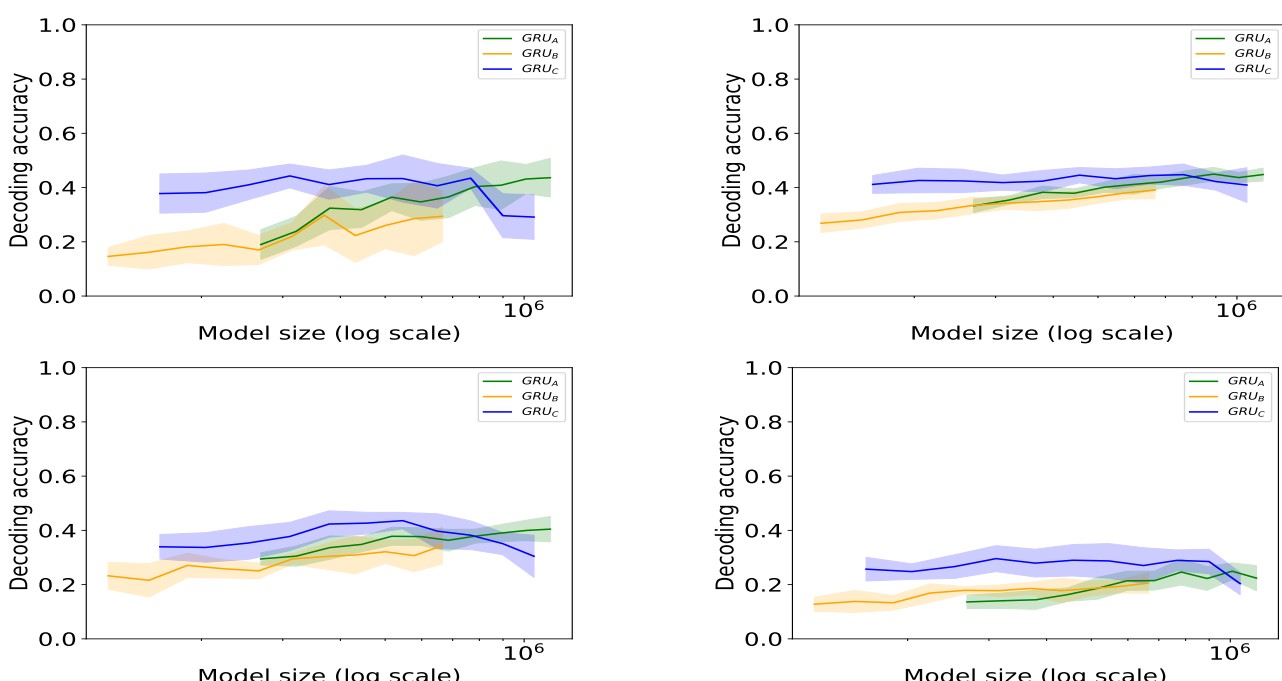

*Figure 6.* Average decoding accuracy = 1 - WER of all four subjects (subject 1 to subject 4, starting from top left in a clockwise manner). Models are evaluated over 10 random seeds. *Higher is better.*

## B. Training paradigms

We trained our models on the training set and validated them on the validation set. For testing on the test set, we selected the model weights corresponding to the epoch with the minimum validation loss, provided the losses in the immediately preceding and succeeding epochs did not exceed more than 20% of the minimum loss. All GRU models were trained for 100 epochs on Data $_{\text{SMALL-VOCAB}}$ and 500 epochs on Data $_{\text{LARGE-VOCAB}}$. For Data $_{\text{NATO-WORDS}}$, GRU$_A$ was trained for 250 epochs, while GRU$_B$ and GRU$_C$ were each trained for 150 epochs. Model training is completed in just a few minutes, whether using a GPU or CPU.

For WER calculation, we compute the Levenshtein distance between the predicted phoneme sequence and all entries in the word corpora (or combinations of them). When multiple entries share the lowest distance with the predicted sequence and the ground truth is among them, we classify it as a ***wrong*** prediction. For instance, the predicted sequence <T-TH-UW-AH-Z-D-EY> could be decoded as either <TUESDAY> or <THURSDAY>, whose phonetic transcriptions are <T-UW-Z-D-IY> and <TH-ER-Z-D-EY>, respectively. In this case, the ground truth text prompt is <THURSDAY>. However, both <TUESDAY> and <THURSDAY> yield the same minimum Levenshtein distance from the predicted sequence within the 67-word corpora.

Since the articulation is silently produced by the individual, there is no ground truth audio to verify the accuracy of the actual articulation (for example, the subject might have started with TUESDAY and then corrected for THURSDAY). Therefore, recognizing the inherent ambiguity and emphasizing the model's ability to closely approximate the intended word, we can classify such predictions as *correct*. We denote word error rate calculated in this manner as WER$^*$. In this case, the best decoding accuracy on Data $_{\text{SMALL-VOCAB}}$ is $91\%$ (WER$^*$ of just 9%, as opposed to 12% with the previous way of calculating WER).

When such predictions are classified as correct, the generation accuracy of Data $_{\text{NATO-WORDS}}$ significantly increases and is summarized in table 8.

### B.1. Beam search algorithm

We use an algorithm that performs beam search decoding solely on the CTC output probabilities, without incorporating any external language models or prior linguistic knowledge. At each timestep, it evaluates the likelihood of extending existing sequences based purely on the symbol probabilities provided by the CTC output, maintaining a fixed number of the most probable beams (defined by beam width), and ultimately returns the most likely sequence based on the CTC probabilities.

*Table 8.* Best decoding accuracy across all GRU models and model sizes of all four subjects (best value is calculated by averaging over 10 random seeds). When multiple entries in the word corpora share the lowest Levenshtein distance with the predicted sequence and the ground truth is among them, we classify it as a ***correct*** prediction. Results are for Data $_{\text{NATO-WORDS}}$.

| Subject | Best accuracy (1 - WER$^*$) |
|---|---|
| 1 | 54.48% |
| 2 | 59.45% |
| 3 | 38.90% |
| 4 | 56.29% |
| **Average** | **52.28%** |

## C. Background on Riemannian geometry of SPD matrices

Speech articulation involves the coordinated activation of various muscles, with their activation patterns defined by the functional connectivity of the underlying neuromuscular system. Consequently, EMG signals collected from multiple, spatially separated muscle locations exhibit a time-varying graph structure. Gowda et al. demonstrate that the graph edge matrices corresponding to orofacial movements underlying speech articulation are inherently distinguishable on the manifold of SPD matrices. Through experiments with 16 subjects, they highlight the effectiveness of using SPD manifolds as an embedding space for these edge matrices. Building on this foundation, we investigate the temporal evolution of graph connectivity using edge matrices to enable EMG-to-language translation.

Directly working with SPD matrices using affine-invariant or log-Euclidean metrics (Arsigny et al., 2007) involves computationally expensive operations, such as matrix exponential and matrix logarithm calculations. These operations make mappings between the manifold space and the tangent space, and vice versa, computationally intensive. To address this, Lin proposed methods to operate on SPD matrices using Cholesky decomposition. They established a diffeomorphism between the Riemannian manifold of SPD matrices and Cholesky space, which was later utilized by Jeong et al. to develop computationally efficient recurrent neural networks. In Cholesky space, the computational burden is significantly reduced: logarithmic and exponential computations are restricted to the diagonal elements of the matrix, making them element-wise operations. Additionally, the Fréchet mean is derived in a closed form.

Given a set of SPD edge matrices $\mathcal{E}(\tau)$ over different time windows $\tau$, we first calculate their corresponding Cholesky decompositions $\mathcal{L}(\tau) = \text{CHOLESKY}(\mathcal{E}(\tau))$, such

that $\mathcal{E}(\tau) = \mathcal{L}(\tau)\mathcal{L}(\tau)^T$. Then, the Fréchet mean of the Cholesky decomposed matrices $\mathcal{L}(\tau)$ is given by

$$\mathcal{F}_{\text{CHOLESKY}} = \frac{1}{n}\sum_{i=1}^{n}\lfloor\mathcal{L}(\tau_i)\rfloor \quad +$$
$$\exp\left(\frac{1}{n}\sum_{i=1}^{n}\log(\mathbb{D}(\mathcal{L}(\tau_i)))\right).$$

The Fréchet mean $\mathcal{F}$ on the manifold of SPD matrices is calculated as

$$\mathcal{F} = \mathcal{F}_{\text{CHOLESKY}}\mathcal{F}_{\text{CHOLESKY}}^T.$$

In the above equation, $\lfloor\mathcal{L}(\tau)\rfloor$ is the strictly lower triangular part of the matrix $\mathcal{L}(\tau)$, and $\mathbb{D}(\mathcal{L}(\tau))$ is the diagonal part of the matrix $\mathcal{L}(\tau)$.

$\text{GRU}_A$ is a standard GRU (Chung et al., 2014). $\text{GRU}_B$ is constructed from $\text{GRU}_A$ by replacing the arithmetic operations of $\text{GRU}_A$ defined in the Euclidean domain with the corresponding operations on the SPD manifold. Gates of $\text{GRU}_B$ as defined by Jeong et al. are given below. Given the sparse SPD edge matrices $\sigma(\tau)$ over different time windows $\tau$, we first calculate their corresponding Cholesky decompositions $l(\tau) = \text{CHOLESKY}(\sigma(\tau))$, such that $\sigma(\tau) = l(\tau)l(\tau)^T$.

Update-gate $z_\tau$ at time-step $\tau$ is

$$z_\tau = \text{SIGMOID}(w_z\lfloor l_\tau\rfloor + u_z\lfloor h_{\tau-1}\rfloor + b_z)+$$
$$\text{SIGMOID}(b_{z'}[\exp(w_{z'}\log(\mathbb{D}(l_\tau))+$$
$$u_{z'}\log(\mathbb{D}(h_{\tau-1}))]). \quad (1)$$

Reset-gate $r_\tau$ at time-step $\tau$ is

$$r_\tau = \text{SIGMOID}(w_r\lfloor l_\tau\rfloor + u_r\lfloor h_{\tau-1}\rfloor + b_r)+$$
$$\text{SIGMOID}(b_{r'}[\exp(w_{r'}\log(\mathbb{D}(l_\tau))+$$
$$u_{r'}\log(\mathbb{D}(h_{\tau-1}))]). \quad (2)$$

Candidate-activation vector $\hat{h}_\tau$ is

$$\hat{h}_\tau = \text{TANH}(w_h\lfloor l_\tau\rfloor + u_h(\lfloor r_\tau\rfloor * \lfloor h_{\tau-1}\rfloor) + b_h)+$$
$$\text{SOFTPLUS}(b_{h'}\exp(w_{h'}\log(\mathbb{D}(l_\tau))$$
$$+ u_{h'}\log(\mathbb{D}(r_\tau) * \mathbb{D}(h_{\tau-1})))). \quad (3)$$

Output vector $h_\tau$ is

$$h_\tau = (1 - \lfloor z_\tau\rfloor) * \lfloor h_{\tau-1}\rfloor + \lfloor z_\tau\rfloor * \lfloor\hat{h}_\tau\rfloor+$$
$$\exp((1 - \mathbb{D}(z_\tau)) * \log(\mathbb{D}(h_{\tau-1}))+$$
$$\mathbb{D}(z_\tau) * \log(\mathbb{D}(\hat{h}_\tau))). \quad (4)$$

In the above equations, $h_{\tau-1}$ is the hidden-state at time-step $\tau - 1$.

In $\text{GRU}_C$, we define an additional implict layer solved using neural ODEs. The dynamics $f$ of EMG data is modeled by a neural network with parameters $\Theta$. The output state $h_\tau$ is updated as,

$$h_{\tau-1} \leftarrow \text{ODESOLVE}(f_\Theta, \widetilde{\text{LOG}}(h_{\tau-1}), (\tau - 1, \tau))$$
$$h_\tau = \text{GRU}(l_\tau, \widetilde{\text{EXP}}(h_{\tau-1})), \quad (5)$$

where $\widetilde{\text{LOG}}$ is the logarithm mapping from the manifold space of SPD matrices to its tangent space and $\widetilde{\text{EXP}}$ is its inverse operation as defined by Lin. GRU is a gated recurrent unit whose gates are given by equations 1 - 4.

Previous work by Gowda & Miller demonstrated the effectiveness of SPD matrices in decoding *discrete* hand gestures from EMG signals collected from the upper limb. Furthermore, SPD matrix representations have been extensively utilized to model electroencephalogram (EEG) signals, although they have never been applied to complex tasks such as sequence-to-sequence speech decoding. For example, Barachant et al.; Barachant et al. employed Riemannian geometry frameworks for classification tasks in EEG-based brain-computer interfaces, while Sabbagh et al. developed regression models based on Riemannian geometry for biomarker exploration using EEG data.

The novelty of our work lies in the algebraic interpretation of manifold-valued data through linear transformations, and the development of models for complex sequence-to-sequence tasks. This approach moves beyond the conventional applications of classification and regression.

## D. $\sigma(\tau)$ are sparse matrices

We show that $\sigma(\tau)$ are indeed sparse matrices in figure 7.

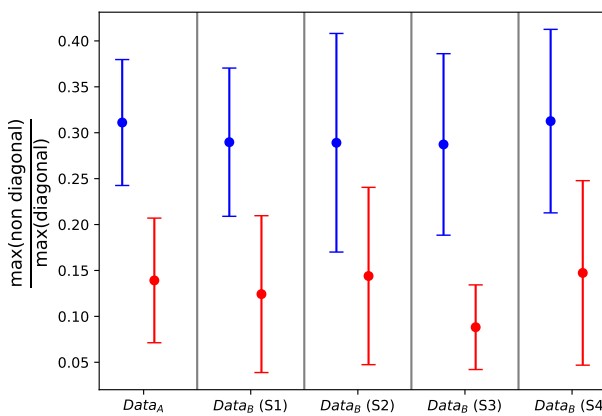

*Figure 7.* Blue: Average value of $\frac{\max(\text{ABS}((\text{NON DIAG}(\Sigma(\tau))))}{\max(\text{DIAG}(\Sigma(\tau)))}$ for all $\tau$ in train-validation-test set. Red: Average value of $\frac{\max(\text{ABS}((\text{NON DIAG}(\sigma(\tau)))}{\max(\text{DIAG}(\sigma(\tau)))}$ for all $\tau$ in train-validation-test set. As we can see, $\sigma(\tau)$ are approximately diagonal compared to $\Sigma(\tau)$. We use sparse matrices $\sigma(\tau)$ for EMG-to-language translation. Subjetcs are abbreviated with notation *S1, S2, S3, S4*.

## E. Effect of training data size on decoding accuracy

We train the GRU$_A$ model with varying train dataset sizes for Data $_{\text{SMALL-VOCAB}}$ and present the decoding accuracy and Levenshtein distance in figure 8. As we can see, decoding accuracy demonstrates a plateauing trend with increase in train dataset size, but importantly, has not saturated yet. In future, we would like to explore if more training data can lead to better decoding accuracy.

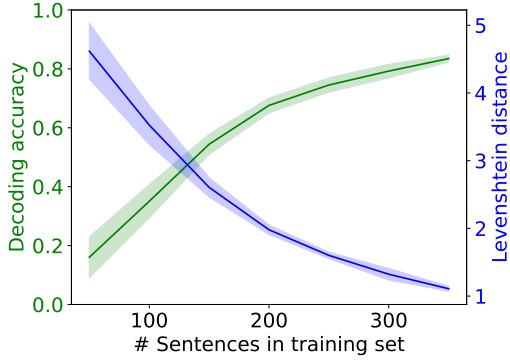

*Figure 8.* Decoding accuracy and Levenshtein distance versus training dataset size for Data $_{\text{SMALL-VOCAB}}$. All experiment parameters are same as in table 1 except for the varying train set size. We use GRU$_A$ for training.

## F. Electrode position versus decoding accuracy

The form factor of an EMG-based neuroprosthesis plays a critical role in its usability, particularly in facilitating ease of application and removal. Here, we evaluate three electrode configurations. We have 31 electrodes placed on the throat, neck and the left cheek. The configurations are defined as follows:

① Configuration$_A$: We consider electrodes placed on the throat and left neck only (10 electrodes).
② Configuration$_B$: We consider electrodes placed on the left cheek, with the neck electrodes excluded (11 electrodes).
④ Configuration$_C$: We consider electrodes on the throat and left neck and cheek, excluding those on the right neck (22 electrodes).

This exploration aims to assess the practicality and performance of each configuration to inform design choices for an optimal neuroprosthetic interface. Electrode placement on the throat, the left neck and left cheek is same as described in Gowda et al. Electrode placement on the right neck is symmetrical to that of left neck. Decoding accuracy for various configurations are shown in table 9. The training paradigm is same as in table 1, except for the varying number of electrodes. Decoding accuracy are obtained using GRU$_A$ for Data $_{\text{SMALL-VOCAB}}$.

*Table 9.* Decoding accuracy with various electrode configurations.

| Configuration | Accuracy (1 - WER) |
|---|---|
| Configuration$_A$ | 86.00% |
| Configuration$_B$ | 84.24% |
| Configuration$_C$ | 85.96% |

Above results show that EMG based neuroprosthesis can have a small form factor (such as neck only or cheek only), and still provide good decoding accuracy (decoding accuracy using all 31 electrodes is *88%*).

## G. Text to personalized audio synthesis

The generated personalized audio files will be made available as part of the open-sourced codes.

We synthesize constructed phoneme sequences into personalized audio using methods described by Choi et al. For this, we train the model proposed by Choi et al. on speech corpora provided by Panayotov et al. (LibriSpeech TRAIN-CLEAN-360 and TRAIN-CLEAN-100) and Veaux et al.

825 (VCTK corpus). For few-shot learning, we use a 40-second
826 reference audio clip from the subject (Data $_{\text{SMALL-VOCAB}}$) to
827 capture the speaker's vocal characteristics.

830 The process involves converting the predicted text into audio
831 using Google Text-to-Speech (gTTS). The gTTS-generated
832 audio is then personalized using the model by Choi et al.,
833 leveraging the 40-second reference audio data (the refer-
834 ence audio includes linguistic content $L$ that is absent from
835 the Data $_{\text{SMALL-VOCAB}}$). This approach ensures that the syn-
836 thesized audio closely mimics the speaker's unique vocal
837 attributes.

