# OpenReview forum: "Non-invasive electromyographic speech neuroprosthesis: a geometric perspective"
_ICML.cc/2025/Conference — Submitted to ICML 2025_

### Official Review · Reviewer_cEe2 · 2025-03-08

**Overall Recommendation:** 2

**Summary:**

The authors demonstrate a system that translates silently articulated speech into both text and audio. The method collects electromyogram (EMG) signals from multiple articulatory sites on the face and neck during alaryngeal speech and uses GRU with CTC loss to perform sequence-2-sequence decoding. Experiments are conducted to demonstrate the effectiveness of this work.

**Claims And Evidence:**

Yes, the authors supported their claim by conducting experiments.

**Essential References Not Discussed:**

N/A

**Ethical Review Concerns:**

Human participants are involved.

**Ethical Review Flag:**

Flag this paper for an ethics review.

**Ethics Expertise Needed:**

["Other expertise"]

**Experimental Designs Or Analyses:**

Yes.

**Methods And Evaluation Criteria:**

Yes, both the method and evaluation criteria make sense.

**Other Comments Or Suggestions:**

N/A

**Other Strengths And Weaknesses:**

* a) While the proposed work demonstrates effectiveness in EMG-to-text and EMG-to-audio decoding, its contribution to the representation learning community is limited. The approach is largely built upon existing technologies; for instance, the use of a GRU with CTC loss has already been reported by Willett et al. in their brain language decoding framework, and the language corpora are similarly adopted. Moreover, the manifold of SPD matrices is well explored in prior work by Gowda et al. Overall, the novelty of this work does not appear to align well with the scope of ICML.

* b) The experimental details are insufficient. Given that the proposed approaches aim to address the needs of individuals who have lost intelligible speech due to laryngectomy, neuromuscular disease, stroke, or trauma, it is unclear whether the enrolled participants are healthy subjects or patients. Since muscle movement patterns can differ significantly between healthy individuals and patients, simply simulating silent speech may not accurately capture the EMG patterns in the target population.

* c) Compared to prior work by Gowda et al., the primary difference in this study is the use of silent speech data instead of audible articulation. This difference alone does not represent a substantial contribution.

* d) The paper lacks comparisons with alternative decoding algorithms beyond GRU-based or other state-of-the-art EMG decoding approaches, making it difficult to appreciate the unique contributions of the proposed algorithm.

**Questions For Authors:**

N/A

**Relation To Broader Scientific Literature:**

Limited to the domain of HMI in general.

**Theoretical Claims:**

N/A

---

> ### Author Rebuttal · Authors · 2025-03-27
>
> **Response to comment a)**
> We thank the reviewer for raising this point. Reference [1] presents a high-performance BCI (brain-computer interfaces) system that records neural activity from the motor cortex at single-neuron resolution. Their approach, involving CNNs for feature extraction and RNNs for temporal modeling, aligns with widely-used sequence-to-sequence architectures.
>
> Our approach differs fundamentally due to its focus on *non-invasive* surface EMG. Unlike intracortical recordings, EMG reflects the summation of motor unit action potentials from multiple muscles, resulting in lower-resolution signals with interference. To address this, we introduce SPD matrix representations that encode second-order channel correlations and provide a compact, discriminative signal representation. This forms the basis for sequence-to-sequence modeling in our EMG-to-phoneme translation task.
>
> Our work performs *phoneme-by-phoneme translation* of silently articulated EMG. This represents a substantial leap over prior efforts that focused on isolated word or phoneme classification (such as in [2]) and generalizes to full language corpora. Even though we cannot record activity at the resolution of individual neurons, our results demonstrate that fine-scale speech decoding is still feasible with non-invasive signals with appropriate representation—an important and novel contribution for speech neuroprostheses.
>
> Overall, we believe our contributions are important and substantial for the field of EMG, which is emerging as a new modality alongside *images*, *audio*, and *video* as recently demonstrated by [5].
>
> **Response to comment b)**
> We appreciate the reviewer’s concern. Our study involved healthy volunteers, in line with institutional ethical guidelines and approvals. As with other early-stage BCI research (e.g.,[3] and [5]), healthy subjects are commonly used to demonstrate feasibility. Our goal is to establish that high-fidelity phoneme decoding is possible from surface EMG, paving the way for future clinical applications. We respectfully note that this study serves as a *first proof-of-concept* of such capabilities. Please refer to rebuttals for **reviewers 3dnc, GLiL** for explanation of signal distribution shift across individuals.
>
> **Response to comment c)**
> We respectfully disagree with the reviewer’s interpretation. Reference [2] addresses closed-set classification tasks (e.g., 36 words or 38 phonemes), whereas our work focuses on *open-set phoneme sequence decoding* from continuous EMG spanning general English language corpora. These are fundamentally different problems. Classification selects from a fixed list of categories, while sequence decoding must infer variable-length output sequences aligned with temporally evolving input. We have clearly described this distinction in the manuscript, including the increased complexity and novelty of the decoding task. Therefore, the comment that our work does not differ from [2] is factually incorrect.
>
> **Response to comment d)**
> We have made every effort to compare our work against representative baselines and relevant prior research in both invasive and non-invasive BCI domains. Specifically, we include comparisons with state-of-the-art invasive BCI systems such as [1], as well as recent advances in non-invasive BCI, including [3]. To the best of our knowledge, the only reproducible and publicly available baseline for EMG-based speech neuroprostheses prior to our work is [4], and we have conducted direct comparisons with this approach in our study.
>
> It is important to note that the field of speech BCI - particularly non-invasive EMG-based speech decoding—is still in its early stages, with relatively few prior works and limited established benchmarks. Within this context, we believe our comparisons are both fair and comprehensive, and we have taken care to position our contributions appropriately relative to the current state of the field.
>
>
>
>  [1] Willett, F. R. et al. A high-performance speech neuroprosthesis. *Nature*, 620(7976), 1031–1036, 2023.
>
>  [2] Gowda, H. T., et al. Geometry of orofacial neuromuscular signals: speech articulation decoding using surface electromyography *arXiv preprint* arXiv:2411.02591, 2024.
>
> [3] Defossez, A., et al. Decoding speech perception from non-invasive brain recordings. *Nature Machine Intelligence*, 5(10):1097–1107, 2023.
>
> [4] Gaddy, D. et al. Digital voicing of silent speech. In Proceedings of the 2020 Conference on Empirical Methods in Natural Language Processing *EMNLP*, pp. 5521–5530, 2020
>
> [5] Ctrl-labs at Reality Labs, et al. "A generic noninvasive neuromotor interface for human-computer interaction." *Biorxiv* (2024): 2024-02.

---

### Official Review · Reviewer_GLiL · 2025-03-12

**Overall Recommendation:** 2

**Summary:**

The paper introduces a method for converting EMG signal into phoneme sequences without requiring audible speech. The core idea is to use CTC loss and inference, which obviates the need for explicit alignment between the input EMG signals and the output phoneme sequences. The proposed method leverages the functional connectivity of the EMG signals by modeling them as symmetric positive definite (SPD) matrices on a Riemannian manifold. Recurrent neural networks (GRU variants) take these (normalized) SPD matrices and outputs phoneme probability distributions at each time frame. The proposed approach is evaluated on three datasets, including one where subjects articulate using NATO codes.

**Claims And Evidence:**

* The paper’s central claim is that the use of SPD representations, combined with CTC loss, significantly improves performance in silent speech decoding. However, the experimental results appear to mix the contributions of the CTC loss and the SPD-based feature modeling. The claims would be strengthened if the paper shows ablation studies that isolate the performance gains attributable to SPD vs. CTC.
* While the authors suggest that modeling the relationship of EMG signals on the SPD is advantageous, the empirical evidence supporting benefits of SPD is not entirely convincing – such as cross-subject generalization, invariance to individual differences, investigation of the changes in the adjacency matrix, etc. Additional experiments that explicitly demonstrate the advantages of the unique characteristics of the method would be beneficial.

**Essential References Not Discussed:**

The paper adequately discusses relevant references.

**Experimental Designs Or Analyses:**

* Although I am not an expert in EMG-based models, the experimental design appears to follow standard practices in the domain, particularly those established in previous works.

**Methods And Evaluation Criteria:**

* Proposed method is evaluated on three datasets (Section 4.1-4.3). Although the datasets are relatively small compared to common ML datasets, they are consistent with those used in prior work, thereby supporting the authors’ choice of data.
* One weakness is the limited comparison to prior works. Furthermore, many details are explained in Table captions (ex: Table 4), which impairs readability. It is recommended to put such details into the main text or organize them in a Table.
* The metrics in the paper, “Levenshtein distance”, “WER”, and “decoding accuracy” are used somewhat interchangeably. Please consider unifying the metrics. In my perspective, Levenshtein distance captures the same result as WER (as in speech recognition convention). However, accuracy should not be computed as 1-WER; it should be measured as binary decision per utterance/word: either entirely correct (1) or wrong (0). Additionally, please consider reporting both PER and WER for all experiments for comprehensive understanding.
* Figure 8, which shows inverse proportional results between decoding accuracy and Levenshtein distance, seems somewhat trivial.

**Other Comments Or Suggestions:**

* The paper focuses on EMG-to-phoneme translation, rather than generating the final speech itself. Text-to-personalized audio synthesis relies on the off-the-shelf models (Appendix G). Clarifying the scope of the paper would make its contributions clearer.
* Please format the references to include publication years – for example, “Gaddy & Klein; Gaddy & Klein” seems to be repeated twice, despite referring to different papers.
* Based on the results in Figures 3, 4, and 6, model scale-up does not bring noticeable performance improvement. Can you add some explanation on this observation?

**Other Strengths And Weaknesses:**

Please see other sections.

**Questions For Authors:**

* In speech recognition, CTC decoding often utilizes a beam width of 50–100. Have the authors experimented with larger beam widths, and if so, what was the impact on performance?
* It seems that each trained model is only applicable to the specific person whose data is used during training. Is this a common approach in the domain or is it a limitation of this work?
* Although not directly related, the problem and the proposed approach seems to be related to the sparse sensing literature. It would be interesting to find/discuss the relationship between two.

**Relation To Broader Scientific Literature:**

The paper is closely related to speech recognition literature, especially using CTC for alignment-free phoneme recognition.

**Theoretical Claims:**

* The paper utilizes established components, such as GRU variants and SPD matrices, to model the inter-sensor relationships. Both have been discussed/proposed in previous studies, and details are properly explained in Appendix. The novelty of the paper lies in its integration of these components, including modifications such as the use of an approximated common eigen basis, computation of the Fréchet mean on the training set, and the application of CTC-based training and inference. These methodological adaptations are sound and well-motivated.

---

> ### Author Rebuttal · Authors · 2025-03-29
>
> **Response to claims and evidence**
>
> *Comment 1)* We thank the reviewer for their comment and the opportunity to clarify. It is important to note that the CTC loss and the SPD matrix representation in our pipeline serve fundamentally different purposes. The use of CTC enables us to demonstrate that direct phoneme sequence translation from silently articulated EMG ($E_S$) is feasible—without relying on aligned EMG ($E_A$) or audio ($A$). This represents a significant advancement in the modeling paradigm itself.
>
> On the other hand, the SPD matrix representation is employed during feature extraction, as it effectively captures the structure of EMG signals in a way that reflects articulatory dynamics. As demonstrated in [2], such representations are well-suited to distinguishing subtle orofacial movements involved in speech, including variations in tongue and jaw positions. These articulatory features are essential for phoneme production, and SPD matrices allow us to model them in a geometry-aware manner. For example, on the manifold of SPD matrices, unsupervised methods like *k*-medoids clustering can naturally distinguish distinct articulatory patterns.
>
> Given the nature of the task, the CTC loss and SPD feature representation operate at different stages of the model and are tightly coupled in the end-to-end framework. Therefore, it is not possible to decouple these components for ablation in isolation.
>
> *Comment 2)* As shown in [2], signal distribution shift across individuals is a covariate shift. This shift can be captured by SPD matrices and can be interpreted as the change of basis. While we mention this in the *discussion section*, the central claim of the paper is that SPD matrix representation can lead to efficient design of neural networks. We support that central claim by designing a single layer recurrent network, and demonstrate 2.4x improvement in WER using a model that is 25x smaller on a limited vocabulary corpora compared to [4].
>
> **Response to methods and evaluation criteria**
>
> *Comment on limited comparison:* Please see the response to **reviewer cEe2**.
>
> *Comment on measurement metrics:*
>  - Levenshtein distance and PER are the same and are used interchangeably.
>  - Accuracy = 1 - WER. They are indeed calculated as binary decision per utterance/word as the reviewer has mentioned.
>
> *Comment on figure 8:* In figure 8, Levenshtein distance is PER and decoding accuracy is 1 - WER and they are not trivially related.
>
> **Response to other comments or suggestions**
>
> In the revised version, we will further clarify regarding EMG to audio. We acknowledge that we use off-the shelf models as explained in Appendix G. We will prominently articulate this in the main text. We will also reformat the citations.
>
> Please see the response to **reviewer 3dnc** regarding model scaling.
>
> **Response to question for authors**
>
> 1) Yes, we have explored decoding with higher beam widths. The performance saturates around the beam width of 5. For example, on the large language corpora, we observe 2 percent points improvement in PER between beam widths 1 and 20 (with most improvement happening when we go from 1 to 5, and eventually plateauing).
>
> 2) Zero-shot and few-shot decoding techniques are not explored in the field of speech EMG, with our work being one of the first works. However, in a related field of decoding hand gestures using EMG, there are some works which demonstrate zero-shot learning (model trained on a set of subjects generalizes to unseen subjects) such as in [5]. In [5], they show that EMG trained on subject $A$ does not generalize to subject $B$. However, EMG trained on thousands of subjects show some generalization capability, but, still requires personalization.
> But [6] argues that pretrained models like in [5] are not of much use and hand gestures can be decoded using EMG in an unsupervised manner on the manifold of SPD matrices using simple $k-$medoids clustering algorithm. Theoretically, EMG can be interpreted as being defined by a set of orthogonal axes that span the space $\mathbb{R}^{|\mathcal{V}|}$, and since we can have infinite such orthogonal bases, generalization across subjects is difficult. It arises due to the purely additive nature of EMG signals.
> As such, we believe that this is not the limitation of our work.
>
> 3) We appreciate the reviewer’s insightful suggestion and will consider investigating the relationship between sparse sensing and our approach in future work.
>
> [6] Gowda, H. T., and Lee M. Miller. "Topology of surface electromyogram signals: hand gesture decoding on Riemannian manifolds." *Journal of Neural Engineering* 21.3 (2024): 036047.

---

### Official Review · Reviewer_3dnc · 2025-03-14

**Overall Recommendation:** 3

**Summary:**

This paper introduces a non-invasive EMG decoder that can produce text or audio from silent speech, without EMG responses from any audible speech. The authors found an efficient sparse decomposition of the responses by analyzing their geometry, and GRUs operating on this manifold significantly improved word error rates.

**Claims And Evidence:**

1. The authors claim "EMG-to-audio" decoding by first decoding a sequence of phonemes, then personalizing the output of an off-the-shelf synthesizer to the specific subject. Since the authors claim to create more than just a "EMG-to-phonemes" decoder, there should be some evaluation about the faithfulness of the speech synthesis system as well.

2. The authors claim:
    > modeling dynamics of EMG signals using neural ODEs is beneficial and allows for better abstraction of the data.

    It's true that GRU_C outperforms the other GRUs at lower model sizes for the Nato and Small-Vocab datasets. But, seeing as the other GRUs end up outperforming GRU_C at larger sizes and are more stable, does this claim still hold? It's not (please correct if this is a wrong assumption) that the larger model size makes the models significantly harder to train, so I'm not convinced that the neural ODE is really helping much.

**Essential References Not Discussed:**

I'm not aware of any missing references.

**Experimental Designs Or Analyses:**

The setup for the experiments and 3 datasets was sound.

**Methods And Evaluation Criteria:**

The evaluation methods match previous literature cited by the authors (Gaddy & Klein, 2020; Willet et al.) to allow for near-direct comparison. Data_Large is actually _more_ challenging than in Willet 2023 because the speech rate is faster.

I have one concern about how the final results are shown in the tables. There are 2 parameters the authors are sweeping along: GRU architecture and hidden unit size. It seems that they are taking the minimum _test_ performance along both dimensions to report e.g. WER in Table 2. But, the test performance of these models is not monotonic over model size (especially for GRU_C) or strongly ordered over architectures (GRU_C often trades places with A/B). How does one know which architecture/size to pick? Taking the minimum _test_ performance seems like it would introduce test set leakage, but using the validation performance instead would remedy that.

**Other Comments Or Suggestions:**

1. I would suggest adding years to citations, at least where it is ambiguous without clicking on the link. For example, "Gaddy & Klein" can refer to either 2020 or 2021. (Also, there is no year in the bibliography for the 1st Gowda citation.)
2. The caption in Figure 1 was very helpful for understanding this decomposition, maybe even without the graphic.
3. I don't see a reference from the main text to Appendix C -- this would be helpful in Sec. 3.1 or 3.2. This specific section of the appendix covers many aspects, and it would be worth adding a reference in the main text for all these instances.
4. The aspect ratios for the appendix figures do not match the main text. And since they're far from where they're first referenced on page 7, the captions should be more descriptive (e.g. which dataset).

**Other Strengths And Weaknesses:**

Other strengths:
1. The paper is well written, and great care is taken to explain the diagonalization of $\mathscr{E}$.
1. This work removes the need to collect strongly aligned data under multiple conditions (audible vs. silent speech), easing the complexity of data collection.

Other weaknesses:
1. The appendix is somewhat unpolished, and is missing key connections with the main text (see the next section for examples).

**Questions For Authors:**

1. In any of the 3 datasets, is any data collected for the same subject across multiple sittings? If so, would $Q$ need to be re-estimated like it's a new subject, or can you take advantage of previously collected data for that subject?
2. During data collection, are the articulators (tongue, lips, etc.) still moving during silent speech? (I assume yes.)
3. How fast were the GRU variants to train/run? Given the neural ODE, was GRU_C noticeably more expensive?
4. What values are used for the GRU hidden dimensionality? And why are the ranges of model sizes different in Figs. 3/4 vs. 5/6?
5. Please clarify how a single model was chosen for Table 2 and for Table 4 among the model sizes and architectures. (See "Methods And Evaluation Criteria" for details of my concerns.)
6. Appendix G references a "40-second reference audio clip", whereas Sec. 3 paragraph 2 talks about "an audio clip of about 3-5 minutes, not necessarily containing the same phonemic content as L, recorded before their clinical condition". Are these describing the same recording, and if so, why is there a discrepancy in the duration?

**Relation To Broader Scientific Literature:**

This work shows that, under the right representation, EMG can rival many impactful BCIs from recent years that are more invasive.

**Theoretical Claims:**

I read through the claims in the main text.

---

> ### Author Rebuttal · Authors · 2025-03-31
>
> **Response to claims and evidence / Methods and evaluation criteria**
>
> *Comment 1*
> Our speech style conversion module follows the implementation provided in [7]. We trained the models using the publicly available code released by [7]. In the revised version, we will move the content currently in Appendix G into the main text and further clarify that we use the model from [7]. We will also clarify that speech style conversion is not the main contribution of our work; rather, our core contribution lies in the EMG-to-phoneme sequence translation framework.
>
> *Comment 2*
> As shown in [2], EMG signals exhibit structured representations on the manifold of SPD matrices, particularly with respect to articulator placement. For example, distinct orofacial movements underlying speech articulation naturally cluster and can be separated using unsupervised algorithms. Building on this insight, we demonstrate that a single-layer recurrent network is sufficient for EMG-to-phoneme translation, and that model size can be effectively controlled via the GRU hidden dimension.
>
> Our motivation for analyzing phoneme error rate (PER) across model sizes was to test whether performance follows the power law behavior described in [8]. Following standard practice, we plotted these trends on the test set; however, the same trends hold on the validation set as well, and we will include those plots in the revised manuscript.
>
> We observe that both $GRU_A$ and $GRU_B$ exhibit power law scaling, where the error $E$ decreases as a function of model size $N$ or data size $D$. Specifically, for models with limited capacity but sufficient data, performance improves as $E = \frac{c_1}{N^{c_2}}$ (Figures 4 and 6), while for models with sufficient size but limited data, the error follows $E = \frac{c_1}{D^{c_2}}$ (Figure 8). However, the law breaks down and plateaus when bottlenecked by either $N$ or $D$, explaining why simply increasing model size does not lead to noticeable gains, as correctly pointed out by **reviewer GLiL**.
>
> In contrast, $GRU_C$ does not follow this trend. While it performs well at smaller model sizes, its performance degrades as $N$ increases without increasing $D$. We agree with the reviewer that training larger models is not inherently more difficult. We will revise the sentence to: “Although modeling EMG using neural ODEs shows benefits at smaller model sizes, its performance decreases when the model is scaled without increasing the data size.”
>
> For the SMALL-VOCAB dataset, the fitted parameters are:
> $GRU_A$: $c_1 = 0.2539$, $c_2 = 0.4931$
> $GRU_B$: $c_1 = 0.2319$, $c_2 = 0.2939$
>
> Despite the simplicity of our model—where size is governed by a single hyperparameter—we observe that scaling laws still hold, consistent with trends seen in larger language models. Given these scaling patterns, for $GRU_A$ and $GRU_B$, model performance can be predicted reliably from data and model size.
>
> **Response to other comments or suggestions**
>
> We thank the reviewer for the suggestions. We will incorporate them in the revised manuscript.
>
> **Response to questions for authors**
>
> *Comment 1*
> All data in this study were collected in a single session, with sensor electrodes placed only once. In scenarios involving multiple sessions, the matrix $Q$ may need to be re-estimated, as precise electrode placement is critical. Even small shifts can result in signals from different muscle units, leading to distributional shifts. This remains an open question in non-invasive BCIs. In contrast, invasive BCIs benefit from surgically fixed electrodes. An important future direction is to explore whether few-shot learning methods can help adapt to such changes. For further discussion, please see our response to **reviewer GLiL**.
>
> *Comment 2*
> Yes. The articulators were moving during the data collection.
>
> *Comment 3*:
> Yes. $GRU_B$ and $GRU_C$ were slower to train compared to $GRU_A$.
> $GRU_A$: ~2 minutes, converged 100 epochs
> $GRU_B$: ~20 minutes, converged 50 epochs
> $GRU_C$: ~20 minutes, 25 epochs
> (All timings measured on an NVIDIA RTX 4090.)
>
> *Comment 4*:
> For SMALL-VOCAB, we used hidden unit dimensions ranging from 230 to 496. For NATO-WORDS, the range was 78 to 253. This difference is due to the number of electrodes: SMALL-VOCAB used 31 (input: 31×31), and NATO-WORDS used 22 (input: 22×22).
>
> *Comment 5*:
> The GRUs follow the same power law behavior on the LARGE-VOCAB dataset. We trained $GRU_A$ with various hidden sizes (training time: ~30 minutes on RTX 4090), and observed $c_1 = 0.5925$, $c_2 = 0.028$.
>
> *Comment 6*:
> The reported 3–5 minute duration refers to a worst-case scenario. For voice cloning, we used a 40-second clip. We will clarify this in the revised manuscript.
>
> [7] Choi, H.-S., et al. Neural analysis and synthesis: Reconstructing speech from self-supervised representations. Advances in Neural Information Processing Systems, 34:16251–16265, 2021.
> [8] Kaplan, J., et al. Scaling laws for neural language models. arXiv preprint arXiv:2001.08361, 2020.

---

### Decision · Program_Chairs · 2025-05-01

**Decision:**

Reject

**Comment:**

This work proposes a model that converts electromyographic signals of silent speech into text or speech signals by a sequence-to-sequence approach with CTC to map a representation of EMG signals into a sequence of phonemes. Multiple reviewers had concerns about the novelty of the approach; while authors have argued that the approach presented here is novel, the concept of a sequence-to-sequence model from a given modality to another, as well as the methods used, are not of general interest to the ICML audience. The work might be a valuable contribution to the specific application, so I believe perhaps ICML is not the best venue.